# Comprehensive Performance Evaluation System Based on Environmental and Economic Benefits for Optimal Allocation of LID Facilities

**Yiran Bai, Yuhong Li \*, Ruoyu Zhang, Na Zhao and Xiaofan Zeng**

School of Hydropower and Information Engineering, Huazhong University of Science and Technology, Wuhan 430074, China; m201773764@hust.edu.cn (Y.B.); zhangry@hust.edu.cn (R.Z.); na.zhao.2011@hust.edu.cn (N.Z.); zengxiaofan@hust.edu.cn (X.Z.)

**\*** Correspondence: liyuhong@hust.edu.cn; Tel.: +86-138-8610-1077

**Abstract:** In recent years, urban flooding occurred frequently because of extreme rainstorms. Sponge city construction can effectively mitigate urban flooding and improve urban rainwater utilization. Low-impact development (LID) is regarded as a sustainable solution for urban stormwater management. In this project, a comprehensive evaluation system was developed based on environmental and economic benefits using the analytical hierarchy process (AHP) and the Storm Water Management Model (SWMM) of the United States (US) Environmental Protection Agency (EPA). The performance of four LID scenarios with the same locations but different sizes of green roof, permeable pavement, concave greenbelt, and rain garden were analyzed in the Sucheng district of Jiangsu province in China. Results illustrate that the green roof performed best in reducing runoff, while the rain garden performed worst. The LID combination scenario (1) that contained more green roof, permeable pavement, and concave greenbelt facilities, but fewer rain gardens had the better comprehensive performance on the basis of environmental and economic benefits. The combined scenario (2) (LID proportion of maximum construction area was 40%) could also be an alternative. This study provides a guide to optimize LID layouts for sponge city construction, which can also provide optimal selection for other sponge city constructions.

**Keywords:** sponge city construction; LID; comprehensive evaluation system; AHP; SWMM

## 1. Introduction

In recent years, high-speed urbanization led to a rapid increase in the impervious area of the surface, and the natural hydrologic cycle changed greatly [1,2]. Due to these changes, urban flooding and runoff pollution occur frequently, which are also caused by extreme climate events [3,4]. China is a society with severe urban flooding problems, and damages caused by flooding are exponentially increasing [5]. A survey indicated that 62% of Chinese cities suffered from floods, with direct economic losses amounting to $100 billion from 2011 to 2014 [6]. However, as developed countries in North America and Europe previously rapidly urbanized, they faced and addressed stormwater problems earlier than China [7]. Urban drainage systems aim at draining surface runoff out of urban areas. However, traditional urban drainage systems are not able to meet the current requirements for protection from urban flooding [8]. Because of the frequent occurrence of flood disasters in China and the precedents of other countries, it is of great significance for China to propose new suitable policies to deal with these hazards [7].

The Chinese government launched sponge city (SPC) construction to address these challenges [9]. The concept of SPC indicates that a city can function as a sponge, whereby it absorbs, stores, infiltrates,

and purifies stormwater, and releases it for reuse when necessary [9–12]. Sponge city construction aims to realize effective control and utilization of city rainwater; it is a new concept that controls rainwater based on green infrastructure, such as rain gardens, green roofs, and so on [13–15]. The sponge city (pilot) problem was launched under the guidance and support of the Ministry of Housing and Rural–Urban Development (MOHURD), Ministry of Finance (MOF), and Ministry of Water Resources (MWR) at the end of 2014. As of now, there are more than 30 pilot sponge cities set up in China. Although some of them provided benefits to urban environment, a range of challenges are still present, such as technical challenges, physical challenges, and financial challenges [5,9]. On the whole, there are three ways for sponge city construction: (1) protection of the original urban ecosystem, (2) ecological remediation and restoration, and (3) low-impact development (LID) [16]. LID systems are characterized by a series of micro-scale stormwater devices which are near to or located at the source of runoff [2]. These LID techniques infiltrate, retain, and purify stormwater at the source by reducing imperviousness of urban areas [17–19]. They mainly include green roofs, permeable pavements, concave greenbelts, rain gardens, and so on [20].

There are many other urban water terms around the world which are similar to SPC, such as sustainable drainage systems (SuDs), water-sensitive urban design (WSUD), and green stormwater infrastructure (GSI) [2,5,21]. SuDs were developed in the United Kingdom in late 1980s and consisted of a diverse range of technologies used to drain stormwater in a more sustainable manner than conventional solutions. These technologies aim to replicate as closely as possible the natural flow of water and to offset the excess runoff caused by urbanization at source [2,5]. The term WSUD was first used in Australia in the 1990s and underwent several upgrades since then. It mainly aims at the protection and management of the urban water circle [2,5]. GSI is a suite of interventions which comprises artificial and natural materials. These materials utilize vegetation to mitigate surface runoff [21]. GSI projects include natural elements (i.e., wetlands, street trees, and rain gardens) and engineered techniques (i.e., green roofs, bio-retention trenches, and permeable pavements), which mimic the hydrology of the natural landscape [22]. The term GSI is widely used in the stormwater literature, and is synonymous with LID [2]. Bai et al. [23] found that LID facilities based on infiltration performed better in runoff reduction than LID facilities based on storage. It was noted that there are many researches on LID techniques.

Green roofs are building roofs which are covered with vegetation, planted on waterproofing membranes. A green roof has a soil layer on a special drainage mat material that can convey excess rainwater off the building's roof; it is also an important part of sponge city construction [4,21]. Alfredo et al. [24] reported that green roofs can delay peak time, with the reduction rate of peak flow ranging from 30% to 78% compared to conventional roofs. Dietz's [25] experimental results indicated that a green roof can reduce flood volume by 60–70% compared with a traditional roof.

Permeable pavements are filled with gravel, paved with a porous or bituminous concrete, allowing for rainwater to flow through it. In addition to reducing surface runoff, permeable pavements can filter pollutants from rainwater, thereby improving the environment [26]. Permeable pavements are recommended as an effective way of managing runoff from the surface and controlling pollutants in the soil [27,28]. They are usually applied to roads, paths, squares, and parking lots. Qin et al. [8] reported that the distributions of peak flow and volume were statistically different between asphalt catchment runoff and permeable pavement underdrain discharge; peak flow from permeable pavement underdrain was less flashy and tended to show less variation overall compared to asphalt runoff, which usually mirrored spikes in rainfall intensity. Abbott and Comino-Mateos [29] found that, on the permeable pavement of a parking lot, an average of 22% of the runoff left the system during the storm.

Concave greenbelts are a type of bio-retention cell containing vegetation such as green grasses. Therefore, they can provide infiltration, storage, and evaporation of rainwater from surrounding areas [23]. Luan et al. [30] evaluated the simulated effectiveness of surface runoff reduction with three concave greenbelts scenarios with a concave depth of 10 cm and concave greenbelt ratios of 50%, 70%, and 90%. The runoff reduction ranged from 5.2% to 57.3% in all types of rainfall events.

Rain gardens (stormwater gardens) usually refer to extensive vegetated depressions that collect urban runoff from surrounding impervious areas, thereby promoting evapotranspiration and infiltration [21,31]. The plants, such as small trees, wildflowers, and ferns, make excess rainwater feasible to flow into a rain garden. In order to make maintenance requirements desirable, rain gardens are to be located near institutions, such as community centers and schools [21]. Avellaneda et al. [32] showed that the flood hygrograph shifted downward after adding green infrastructure facilities such as green gardens.

To quantify LID benefits, researchers should develop some rainfall-runoff models. The US Environmental Protection Agency (EPA) Storm Water Management Model (SWMM) is a dynamic rainfall-runoff simulation model which is used to simulate the formation of urban runoff, and is also widely used for LID simulation [8,13,20,23]. SWMM was developed in 1971 [33], and the current edition is Version 5.1. SWMM 5.1 set up eight LID controls in the LID module, namely green roofs, permeable pavements, bio-retention cells, rain gardens, roof closure facilities, infiltration trenches, rainwater tanks, and grassed swales [34].

This paper analyzed the hydrologic effects of various LID scenarios in Sucheng district, Suqian city, China. A comprehensive evaluation system based on an analytic hierarchy process (AHP) [13] was proposed to quantify the performance of each LID technique, and we used the SWMM model to obtain simulated results. Three indicators were selected: runoff reduction, peak flow reduction, and economical cost. Finally, we provided an optimal scheme for sponge city construction in the case study. The simulation results can provide useful guidance for the selection of LID techniques, and can also provide technical support for sponge city construction in urban areas. Some suggestions for sponge city construction are given in the conclusions.

## 2. Materials and Methods

In order to evaluate comprehensive benefits of LID techniques, Li et al. [13] proposed a comprehensive method based on three aspects: environmental benefits, economic benefits, and social benefits. Considering that social benefits might be difficult to be estimated accurately because of the limited data, we selected environmental benefits and economic benefits in this research. The evaluation procedure of comprehensive benefits is shown in Figure 1.

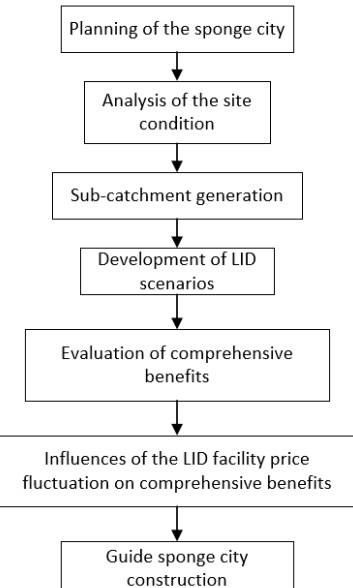

**Figure 1.** The evaluation process of comprehensive performance.

*2.1. Study Area*

The study area was located in the Sucheng district of Suqian city in China, which is surrounded by rivers. It is part of Sucheng district and covers a total area of 270 ha. Sucheng district is the economic center of Suqian city and has an average annual rainfall of 914.9 mm. The rainfall depth can reach 1459.6 mm in wet years. Influenced by a monsoon climate, the rainfall volume is quite unevenly distributed, causing droughts in the spring, autumn, and winter, but frequent rainstorms in the summer [35]. Moreover, the urban area ratio of the study area was more than 80%. Although the densities of the river net are high, flood disasters happen frequently in the study area. During the storm in 17 September 2010, seventeen communities were left without power because of soaked electrical equipment. Direct economic losses amounted to 0.33 billion dollars [36]. During the storm in 23 June 2016, the rainfall depth reached 168 mm within 120 min. Many vehicles were trapped in low-lying areas because of severe waterlogging [37]. Therefore, it is of great significance to design a desirable urban drainage system in Sucheng district to drain excess runoff in time.

The study area was first divided into 16 large-scale sub-catchments according to the rainwater pipe network and land use; each large-scale sub-catchment corresponded to an outfall. Then, the study area was further divided into 55 sub-catchments based on the rainwater pipe network, land use, and Thiessen polygon method. Rivers were treated as several outfalls (Figure 2).

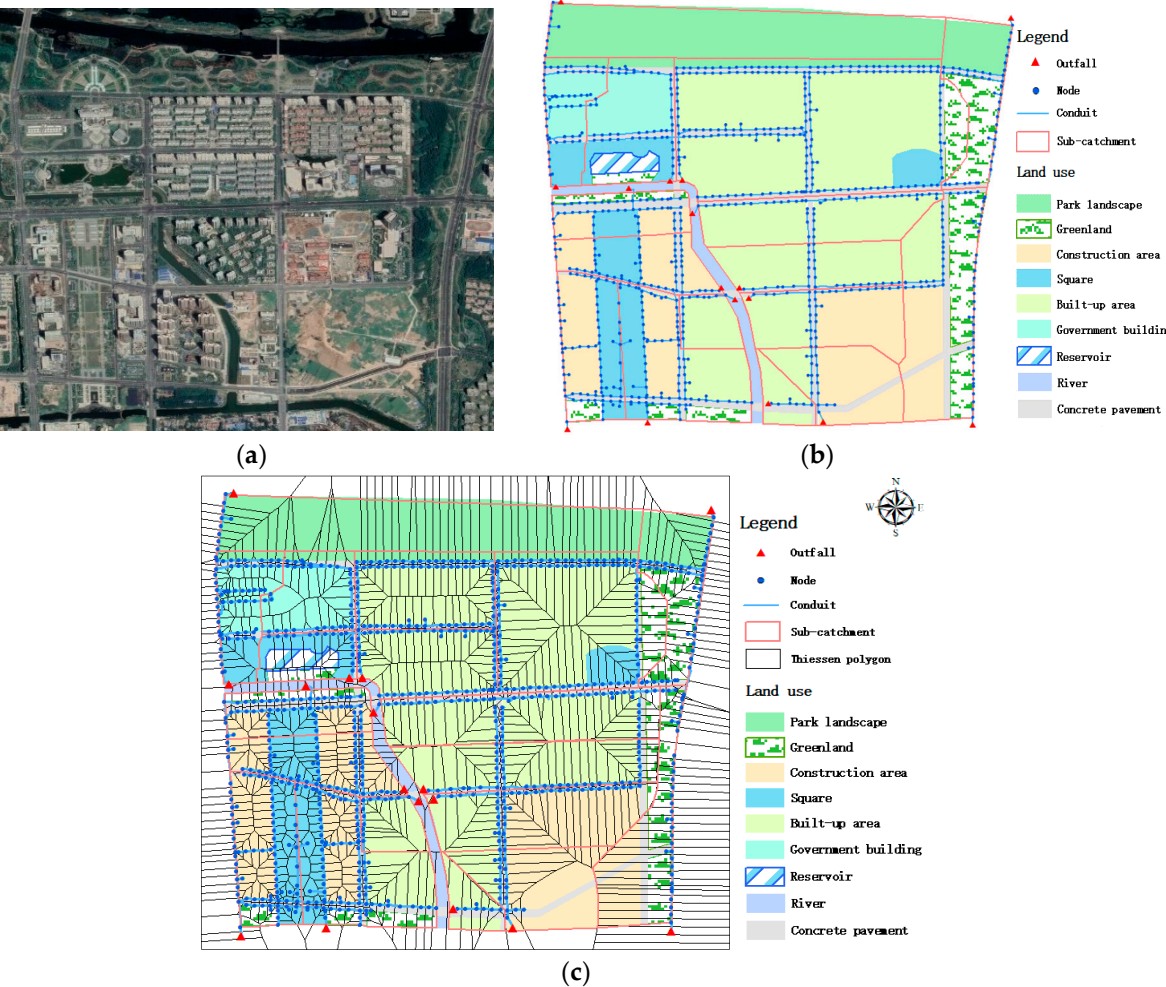

**Figure 2.** Study area of Sucheng district: (**a**) current condition (from Google maps); (**b**) diagram of the pipe network; (**c**) generalized sketch map of sub-catchments. The scale is about 1:30,000.

### 2.2. Determination of Parameters

Water quantity simulation is related to underlying surfaces, and parameters in SWMM are very important to the accuracy of simulation results [20]. Initial water quantity parameters were selected based on empirical values recommend in the SWMM manual. Then, parameters were calibrated and validated based on the comprehensive runoff coefficient (CRC) method [23]. Manning's roughness coefficients for the impervious area (N-Imperv) and the pervious area (N-Perv) were 0.012 and 0.1, respectively; the depth of depression storage on the impervious area (Destore-Imperv) was 3.2 mm, and that on the pervious area (Destore-Perv) was 6.6 mm. We chose the Horton model as the infiltration method. The maximum infiltration rate was 75 mm/h, while the minimum rate was 3.81 mm/h.

We selected these four LID techniques based on the land-use type of the study area, the suggestion of the relevant authority's research [38], and the layout principle of LIDs proposed by Bai et al. [23] and Huang et al. [39]. Green roofs were selected because there are many buildings with traditional roofs in the study area; permeable pavements were selected because there are pedestrian roads, parks, and squares in the study area; rain gardens were selected because there are institutions in the study area; and concave greenbelts were selected because most of the study area comprises low-lying areas. Partial parameters of each LID technique are shown in Tables 1–4.

**Table 1.** Parameters of green roof in Storm Water Management Model (SWMM).

| Surface | Berm Height (mm) | Vegetation Volume Fraction | Surface Roughness | Surface Slope (%) |
|---|---|---|---|---|
| | 50 | 0.2 | 0.13 | 1 |
| Soil | Thickness (mm) | Porosity | Field capacity | Wilting Point |
| | 200 | 0.5 | 0.3 | 0.1 |
| Drainage Mat | Thickness (mm) | Void fraction | Roughness | - |
| | 60 | 0.43 | 0.03 | - |

**Table 2.** Parameters of permeable pavement in SWMM.

| Surface | Berm Height (mm) | Vegetation Volume Fraction | Surface Roughness | Surface Slope (%) |
|---|---|---|---|---|
| | 25 | 0 | 0.12 | 1 |
| Soil | Thickness (mm) | Porosity | Suction Head | - |
| | 150 | 0.5 | 45 | - |
| Pavement | Thickness (mm) | Void Ratio | Permeability (mm/h) | - |
| | 60 | 0.13 | 200 | - |
| Storage | Thickness (mm) | Void Ratio | Seepage rate | Clogging Factor |
| | 250 | 0.43 | 600 | 0 |
| Drain | Flow Coefficient | Flow Exponent | Offset Height (mm) | - |
| | 0.69 | 0.5 | 6 | - |

**Table 3.** Parameters of rain garden in SWMM.

| Surface | Berm Height (mm) | Vegetation Volume Fraction | Surface Roughness | Surface Slope |
|---|---|---|---|---|
| | 150 | 0.1 | 0.12 | 0.3 |
| Soil | Thickness (mm) | Porosity | Conductivity (mm/h) | - |
| | 500 | 0.3 | 500 | - |
| Storage | Thickness (mm) | Void Ratio | Seepage Rate (mm/h) | Clogging Factor |
| | 250 | 0.3 | 400 | 0 |
| Drain | Flow Coefficient | Flow Exponent | Offset Height (mm) | - |
| | 0 | 0.5 | 6 | - |

**Table 4.** Parameters of concave greenbelt in SWMM.

| Surface | Berm Height (mm) | Vegetation Volume Fraction | Surface Roughness | Surface Slope |
|---|---|---|---|---|
| | 150 | 0.05 | 0.12 | 0.1 |
| Soil | Thickness (mm) | Porosity | Conductivity (mm/h) | - |
| | 500 | 0.5 | 110 | - |
| Storage | Seepage Rate (mm/h) | - | - | - |
| | 380 | - | - | - |

## 2.3. Design Scenarios

In order to analyze the hydrological performance of each LID technique and to obtain the optimal proportions of the LID technique, four LID scenarios were designed based on the research of Zhang et al. [40]. As shown in Table 5. Each LID facility was distributed based on the principles proposed by Huang et al. [39] and Bai et al. [23]. Green roofs, concave greenbelts, and rain gardens were deployed in densely populated communities; permeable pavements were deployed on pedestrian roads, parks, and squares; and concave greenbelts and rain gardens are deployed in low-lying parts of the study area. There were 24 sub-scenarios in total because each scenario had six sub-scenarios. In scenario 1, the area of green roofs increased in intervals of 20% from 0% to 100%, while the areas of the other LID facilities occupied 40% of the corresponding maximum construction area. In scenario 2, the area of permeable pavements increased in intervals of 20% from 0% to 100%, while the areas of the other LID facilities occupied 40% of the corresponding maximum construction area. In scenario 3, the area of concave greenbelts increased in intervals of 20% from 0% to 100%, while the areas of the other LID facilities occupied 40% of the corresponding maximum construction area. In scenario 4, the area of rain gardens increased in intervals of 20% from 0% to 100%, while the areas of the other LID facilities occupied 40% of the corresponding maximum construction area. A (83.7742 ha) was the maximum construction area for green roofs, B (45.2075 ha) was the maximum construction area for permeable pavements, C (61.1726 ha) was the maximum construction area for concave greenbelts, and D (50.0771 ha) was the maximum construction area for rain gardens. The proportion of maximum construction area (40%) was determined based on the findings of Zhang et al. [40]. An illustration of the distribution of the four LID facilities for the study area is shown in Figure 3.

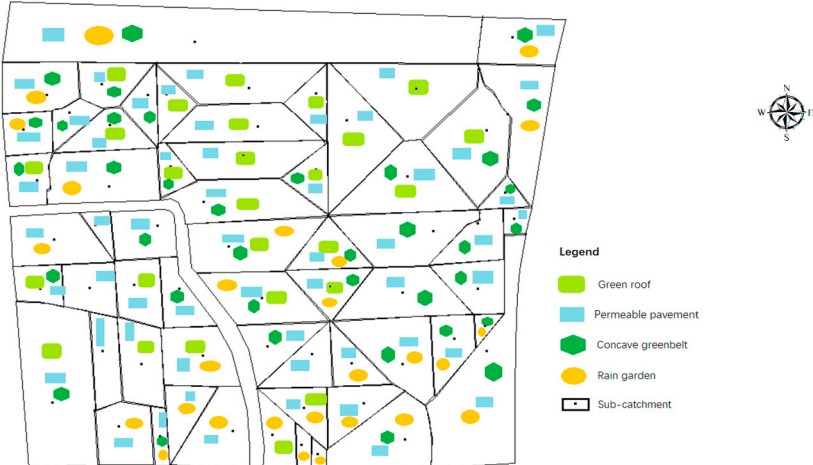

**Figure 3.** The distribution of each low-impact development (LID) facility (the scale is about 1:30,000).

**Table 5.** The proportions of low-impact development (LID) facilities in the four scenarios.

| Scenario | Green Roof | Permeable Pavement | Concave Greenbelt | Rain Garden |
|----------|-----------|--------------------|--------------------|-------------|
| 1 | A* (0%, 20%, 40%, 60%, 80%, 100%) | B* 40% | C* 40% | D* 40% |
| 2 | A* 40% | B* (0%, 20%, 40%, 60%, 80%, 100%) | C* 40% | D* 40% |
| 3 | A* 40% | B* 40% | C* (0%, 20%, 40%, 60%, 80%, 100%) | D* 40% |
| 4 | A* 40% | B* 40% | C* 40% | D* (0%, 20%, 40%, 60%, 80%, 100%) |

### 2.4. Rainfall Simulation

In this research, the Chicago hydrograph model [41] was adopted for the rainfall simulation. The storm events were based on the relationship of rainstorm intensity–duration–frequency in Suqian city, as shown in Equation (1) [42].

$$i = \frac{61.2(1 + 1.05\lg P)}{(t_d + 39.4)^{0.996}}, \tag{1}$$

where $P$ is the return period (year), $t_d$ is the rainfall duration (min), and $i$ is the rainfall intensity (mm/min). The time-to-peak ratio $r$ was set at 0.4. The relationship between rainfall depth and annual total runoff control rate (ATRCR) in Suqian city was examined based on the statistical method of historical rainfall events, from which an ATRCR of about 80% was used for the case of approximately 42 mm of rainfall depth [43]. Therefore, we determined a return period of 80% for the design rainfall, with a rainfall depth of 46.13 mm in 180 min [40]. We chose the rainfall duration of 180 min based on the relevant literature [42]. The design hyetograph developed for the study area from Equation (1) and a return period of 80% is shown in Figure 4.

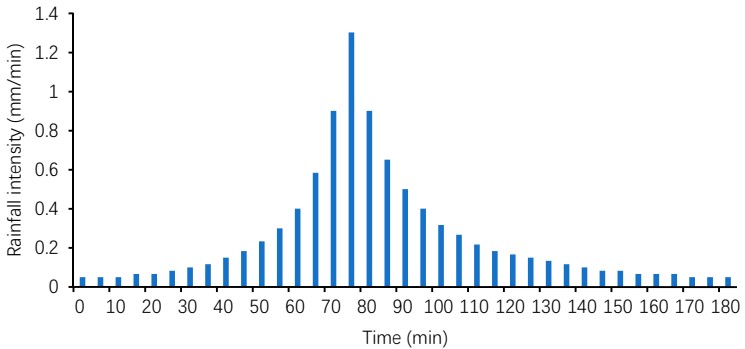

**Figure 4.** Design hyetograph when the return period was taken as 80%.

### 2.5. Design of an Evaluation System Based on AHP Method

AHP was developed by Saaty in the 1970s mainly for multiple-criteria decision problems [44]. It is widely used in fields such as business, education, and industry. In this study, we used AHP to weight selected indicators. We used a weighted summation of indicator values to calculate individual benefits, as described in Equations (2) and (3).

$$B_{im} = \sum_k W_{ki} I_{km}, \tag{2}$$

$$I_{km} = \frac{X_{km}}{\sum_{m=1}^{M} X_{km}}, \tag{3}$$

where $B_{im}$ refers to the benefit $i$ in scenario $m$, $W_{ki}$, refers to the indicator $W_{ki}$'s weight of benefit $i$, $I_{km}$, and $X_{km}$ refer to the normalized value and original value of indicator $k$ in scenario $m$, $M$ refers to the total number of LID scenarios, and $k$ refers to the total number of indicators.

The comprehensive benefit was determined based on the environmental benefit and economic benefit (i.e., economic cost). The values of these indicators were ranked as follows: environmental benefit (0.691) > economic benefit (0.309). The environmental benefit was a positive number, but economic benefit was a negative number. Sub-indicators of environmental benefit were runoff reduction (0.667) and peak reduction (0.333) [6]. The indicator became more important as the weight value increased. Based on regional price-level changes and relevant literature [40,45], the economic costs of typical LID measures in Suqian city were determined, as shown in Table 6.

**Table 6.** Economic costs of typical LID measures.

| LID Type | Green Roof | Permeable Pavement | Concave Greenbelt | Rain Garden |
|---|---|---|---|---|
| Economic cost ($/m$^2$) | 23.30–30.58 | 17.47–32.03 | 6.55–21.84 | 58.24–160.16 |

## 3. Results and Discussion

### 3.1. Environmental Benefits

Environmental benefits were represented by two corresponding sub-indicators of runoff reduction and peak reduction, which were simulated to compare their reduction rates before and after the addition of LID measures. Changes in quantity of runoff (runoff volume reduction, peak reduction) are presented in Tables 7 and 8 and Figure 5.

**Table 7.** Reduction rates of runoff in various LID scenarios.

| LID Proportion of Maximum Construction Area | 0% | 20% | 40% | 60% | 80% | 100% |
|---|---|---|---|---|---|---|
| Scenario 1 | 20.81% | 28.27% | 35.75% | 43.26% | 50.81% | 58.37% |
| D-value * | - | 7.46% | 7.48% | 7.51% | 7.54% | 7.57% |
| Scenario 2 | 28.24% | 31.99% | 35.75% | 39.52% | 43.31% | 47.11% |
| D-value * | - | 3.75% | 3.76% | 3.77% | 3.78% | 3.80% |
| Scenario 3 | 27.75% | 31.74% | 35.75% | 39.78% | 43.83% | 47.91% |
| D-value * | - | 3.99% | 4.01% | 4.03% | 4.05% | 4.08% |
| Scenario 4 | 30.35% | 33.04% | 35.75% | 38.49% | 41.26% | 44.06% |
| D-value * | - | 2.69% | 2.71% | 2.74% | 2.77% | 2.81% |

\* D-value refers to the difference values between 0% and 20%, 20% and 40%, 40% and 60%, 60% and 80%, and 80% and 100% in each scenario.

**Table 8.** Reduction rates of peak flow in various LID scenarios.

| LID Proportion of Maximum Construction Area | 0% | 20% | 40% | 60% | 80% | 100% |
|---|---|---|---|---|---|---|
| Scenario 1 | 19.00% | 25.39% | 32.13% | 39.26% | 47.01% | 55.54% |
| D-value * | - | 6.39% | 6.74% | 7.13% | 7.75% | 8.52% |
| Scenario 2 | 24.80% | 28.47% | 32.13% | 35.91% | 39.82% | 43.97% |
| D-value * | - | 3.67% | 3.67% | 3.77% | 3.91% | 4.16% |
| Scenario 3 | 25.15% | 28.64% | 32.13% | 35.73% | 39.47% | 43.38% |
| D-value * | - | 3.49% | 3.49% | 3.60% | 3.74% | 3.91% |
| Scenario 4 | 27.07% | 29.55% | 32.13% | 34.79% | 37.51% | 40.34% |
| D-value * | - | 2.48% | 2.58% | 2.65% | 2.72% | 2.83% |

\* D-value refers to the difference values between 0% and 20%, 20% and 40%, 40% and 60%, 60% and 80%, and 80% and 100% in each scenario.

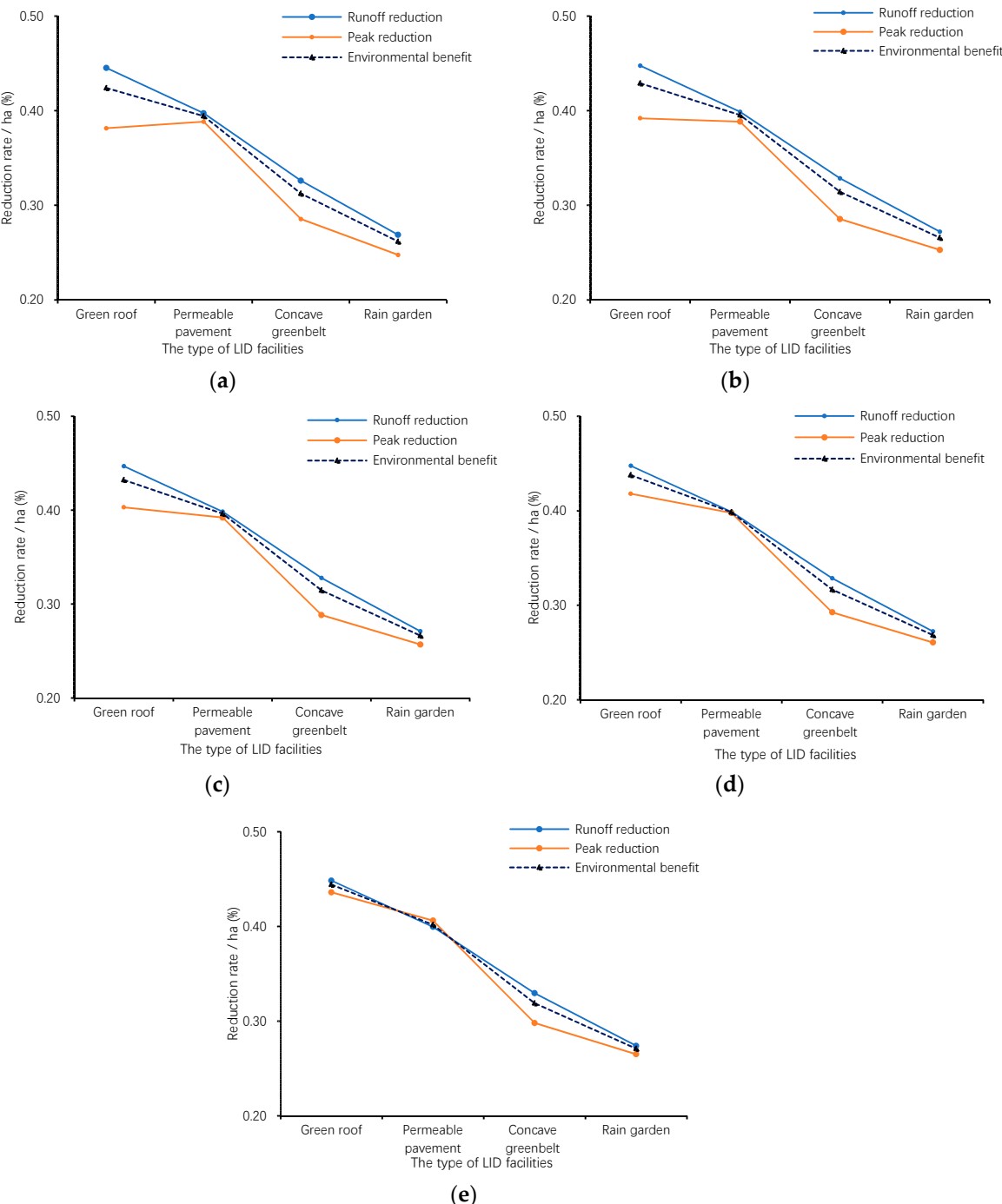

**Figure 5.** Hydrological effectiveness of each LID technique per unit area (note: environmental benefit = 0.667 × runoff reduction + 0.333 × peak reduction). (**a**) Runoff reduction rates of the LID technique when the LID proportion of maximum construction area was set at 20% (column 3 of Tables 7 and 8); (**b**) runoff reduction rates of the LID technique when the LID proportion of maximum construction area was set at 40% (column 4 of Tables 7 and 8); (**c**) runoff reduction rates of the LID technique when the LID proportion of maximum construction area was set at 60% (column 5 of Tables 7 and 8); (**d**) runoff reduction rates of the LID technique when the LID proportion of maximum construction area was set at 80% (column 6 of Tables 7 and 8); (**e**) runoff reduction rates of the LID technique when the LID proportion of maximum construction area was set at 100% (column 7 of Tables 7 and 8).

It was obvious that the reduction rates increased with the increase of the LID proportion of maximum construction area, and the difference values varied between various proportions of the LID

facilities. For scenario 1 with varying proportions of green roofs, the reduction rates of runoff volume and peak flow changed from 20.81% and 19.00% to 58.37% and 55.54%. For scenario 2 with varying proportions of permeable pavements, the reduction rates of runoff volume and peak flow changed from 28.24% and 24.80% to 47.11% and 43.97%. For scenario 3 with varying proportions of concave greenbelts, the reduction rates of runoff volume and peak flow changed from 27.75% and 25.15% to 47.91% and 43.38%. For scenario 4 with varying proportions of rain gardens, the reduction rates of runoff volume and peak flow changed from 30.35% and 27.07% to 44.06% and 40.34%. Taking scenario 1 as an example, the difference values for each proportion of green roofs in runoff reduction were 7.46%, 7.48%, 7.51%, 7.54%, and 7.57%, and those in peak reduction were 6.39%, 6.74%, 7.13%, 7.15%, and 8.52%. The results presented that reduction rates of runoff and peak flow increased more rapidly with the increasing surface area of the LID facility. The reduction rate ranged from 0.27% to 0.45% for runoff volume, and from 0.27% to 0.44% for flow peak.

In general, the reduction rates of LID facilities per unit area (ha) were ranked as follows: green roof > permeable pavement > concave greenbelt > rain garden. Li et al. [13] obtained similar results in their research, whereby concave greenbelts performed better in runoff reduction than rain gardens. In this paper, despite concave greenbelts performing better in runoff reduction than rain gardens, the difference values of runoff reduction rates between concave greenbelts and rain gardens were smaller with the decreasing LID proportion of maximum construction area. The reason might be that the hydrological efficiency of LID measures was influenced by the LID proportion of maximum construction area. Liao et al. [7] thought that permeable pavements performed better than green roofs in runoff and peak reduction, with a permeable pavement area of 2.85 ha and a green roof area of 0.86 ha. As shown in Figure 5, with the decrease in LID proportion of maximum construction area, permeable pavements gradually performed better than green roofs in peak reduction. Li et al. [46] found that permeable pavements with an area of about 0.57 ha performed poorly with regard to the peak flow, but performed well in reducing runoff volume. In this project, permeable pavements performed better in runoff reduction than peak reduction when the LID proportion of maximum construction area was less than 80%. When the LID proportion of maximum construction area was 100%, permeable pavements performed better in peak reduction than runoff reduction. The reason is that the area of LID measures influences the hydrological efficiency.

*3.2. Economic Costs*

Costs of LID facilities were determined based on the local price level and the researches of Zhang et al. [40] and Tang et al. [45]. We selected the typical price of LID facilities as follows: $25.48 per m$^2$ for green roofs, $21.84 per m$^2$ for permeable pavements, $7.28 per m$^2$ for concave greenbelts, and $65.52 per m$^2$ for rain gardens. Calculated values are shown in Table 9.

**Table 9.** Costs of four LID scenarios.

| Scenario | $ (million) | $ (million) | $ (million) | $ (million) | $ (million) | $ (million) |
|---|---|---|---|---|---|---|
| 1 | 19.07 | 23.30 | 27.52 | 31.89 | 36.11 | 40.33 |
| 2 | 23.44 | 25.48 | 27.52 | 29.70 | 31.74 | 33.78 |
| 3 | 21.69 | 26.64 | 27.52 | 28.39 | 29.41 | 30.28 |
| 4 | 14.41 | 20.97 | 27.52 | 34.07 | 40.62 | 47.32 |

*3.3. Comprehensive Benefits of Various LID Scenarios*

The evaluation of comprehensive benefits depended on the individual benefits, the evaluation process of which is summarized in Figure 1. The objective function was given by Equation (2). The objective function values of different price conditions in various LID scenarios are shown in Figure 6.

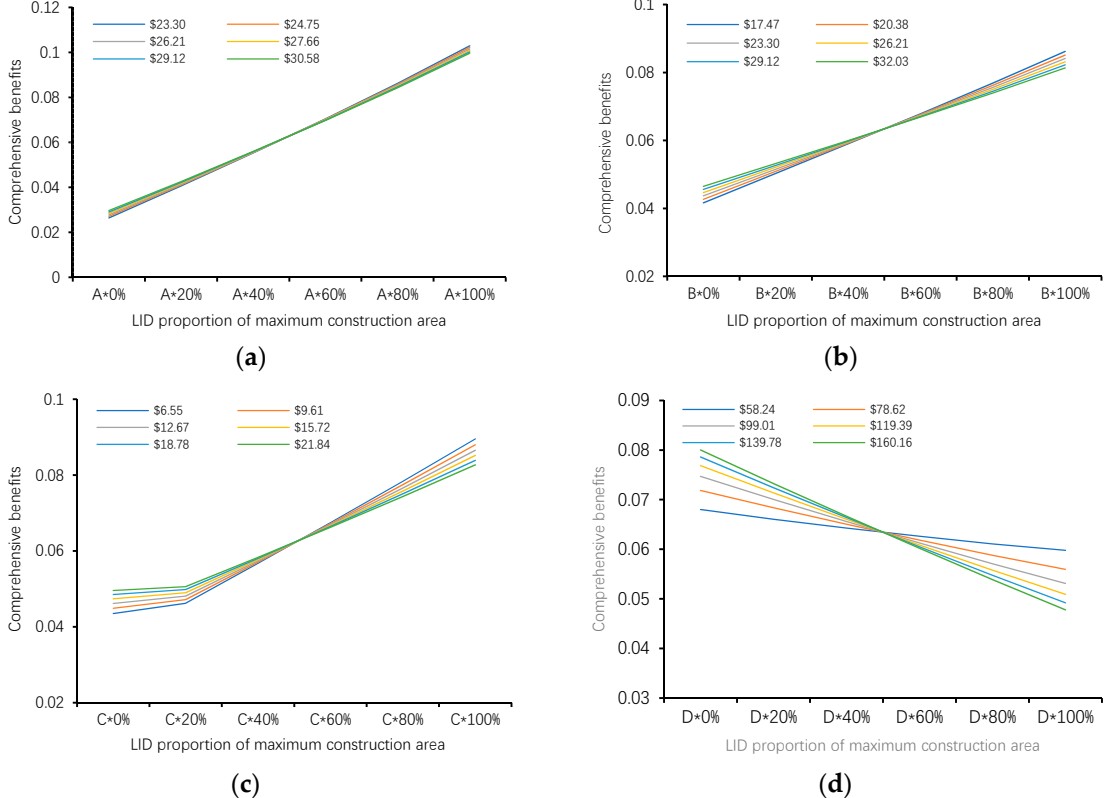

**Figure 6.** Comparison objective function values of different price conditions in various LID scenarios. (**a**) Comprehensive benefits of different price conditions of green roofs in scenario 1; (**b**) comprehensive benefits of different price conditions of permeable pavements in scenario 2; (**c**) comprehensive benefits of different price conditions of concave greenbelts in scenario 3; (**d**) comprehensive benefits of different price conditions of rain gardens in scenario 4.

The weighted summation of normalized indicator values was used to calculate the comprehensive benefits. While the objective function values became bigger, the comprehensive benefits became greater. In scenario 1, the comprehensive benefits ranged from 0.026 to 0.103, while the prices of green roofs ranged from 23.30 ($/m$^2$) to 30.58 ($/m$^2$). In scenario 2, the comprehensive benefits ranged from 0.05 to 0.086, while the prices of permeable pavements ranged from 17.47 ($/m$^2$) to 32.03 ($/m$^2$). In scenario 3, the comprehensive benefits ranged from 0.044 to 0.089, while the prices of concave greenbelts ranged from 6.55 ($/m$^2$) to 21.84 ($/m$^2$). In scenario 4, the comprehensive benefits ranged from 0.048 to 0.073, while the prices of rain gardens ranged from 58.24 ($/m$^2$) to 160.16 ($/m$^2$). For scenario 1, scenario 2, and scenario 3, the comprehensive benefits increased with the increasing area of LID facilities, but the increase rate was reduced with the increase of LID facility prices. For scenario 4, the comprehensive benefits decreased with the increasing area of LID facilities, but the decrease rate increased with the increase of LID facility prices.

In general, in order to provide maximum comprehensive benefits for sponge city construction, combined scenario 1 was considered the top priority, where the green roof facility, permeable pavement facility, and concave greenbelt facility accounted for 100% of the maximum construction area of corresponding LID facilities, and the rain garden facility accounted for 0%. Since combined scenario 2 (A* 40%, B* 40%, C* 40%, D* 40) had a relatively smaller comprehensive benefit, but, at the same time, a smaller area than that of combined scenario 1, the results indicated that combined scenario 2 could be an alternative. The LID proportion of maximum construction area of about 40% was determined based on the experience of Zhang et al. [40]. In addition, all the comprehensive benefits of LID scenarios were positive numbers. According to the LID proportion of maximum construction area in 24 scenarios, the

results suggested that the comprehensive benefits would improve with a larger area of green roofs, permeable pavements, and concave greenbelts, but a smaller area of rain gardens.

## 4. Conclusions

The utilization of LID measures is increasingly important for sponge city construction. To provide a reasonable layout of LID measures in Sucheng district, a comprehensive evaluation system was proposed to quantify the performance of different LID scenarios with regard to the environment and economy. Based on the results of this project, several conclusions can be made as follows:

(1) For environmental benefits, results show that runoff reduction varies with different LID facilities; the green roof performed best, while the rain garden performed worst. The performance of each LID facility in runoff and peak reduction varied with the LID proportion of maximum construction area. As the LID proportion of maximum construction area became smaller, the capacity for stormwater management of each LID facility was limited. Specifically, with the increase of the LID proportion of maximum construction area, green roofs performed better than permeable pavements in reducing runoff volume and peak flow, and permeable pavements performed better in peak reduction than runoff reduction. Compared with the results of Liao et al. [7] and Li et al. [46], the influence of LID proportion of maximum construction area on hydrological performance was evaluated. Overall, LID facilities performed better in the reduction of runoff volume than peak flow.

(2) For economic costs, the unit cost of each LID facility was ranked as follows: rain garden > green roof > permeable pavement > concave greenbelt. The rain garden was the most expensive, while the concave greenbelt was the cheapest.

(3) The comprehensive benefits presented various trends with different proportions of LID facilities. Increased surface area of green roofs, permeable pavements, and concave greenbelts, with fewer rain gardens are suggested for maximum comprehensive benefits; however, combined scenario 2 (A* 40%, B* 40%, C* 40%, D* 40) could also be an alternative. These results can be the reference for the optimization of LID facilities in sponge city construction in Sucheng district. Sponge city construction in the study area can achieve desirable comprehensive performance based on the proposed evaluation system. However, more detailed modeling studies should be taken into account before practical uses of these results, so as to apply LID schemes effectively. In this case, some neglected factors, such as the social factor and water quality factor, might lead to limitations in evaluating the comprehensive benefits of each LID scenario. Despite the limitations, the conclusions are valuable for sponge city construction. For future research, we can carry out a systematic study based on a more integrated methodology, as well as in accordance with specific conditions and actual planning of a sponge city.

**Author Contributions:** Y.B. designed and conducted the experiments. R.Z. and N.Z. collected the data and analyzed the data. Y.B. wrote the draft. Y.L. and X.Z. provided useful advice and checked the paper for revisions and grammar.

**Funding:** This research was financially supported by the National Key Research and Development Program (2016YFC0401005, 2016YFC0401004), and the National Natural Science Foundation of China (91547208, 61841302, 11771449).

**Acknowledgments:** The authors are very grateful to the editor and three reviewers for their very insightful comments and constructive suggestions, which improved the manuscript greatly.

**Conflicts of Interest:** The authors declare no conflicts of interest.

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
