# Peer review of "Comprehensive Performance Evaluation System Based on Environmental and Economic Benefits for Optimal Allocation of LID Facilities"

_water, doi:10.3390/w11020341_

Round 1
Reviewer 1 Report
Worldwide, urban flooding is one of the most frequent and hazardous disasters that can cause enormous impacts on the economy, environment, city infrastructure and human society. In addition, the traditional urban stormwater drainage systems are not able to meet so that to protect the people, environment, etc. Specifically in China, 641 cities of the current 654 face frequent flooding. In 2010, more than 250 cities were affected with a total direct damage of more than 46.9 billion Euro, while more than 230 cities were affected by flooding in 2013. An urban water management program called Sponge City (SPC) is put forward in China in 2014 in order to relieve the flood inundation and water shortage situation. The manuscript investigates the planning of a sponge city in the Sucheng district of Suquian city in China by using the Analytical Hierarchy Method (AHM) and the Storm Water Management Model (SWMM) of USEPA. In this way, the manuscript investigates an important environmental topic with global interest. The methodology used could be developed into a useful tool. However, this manuscript needs an important effort in order to be suitable for publish.
General remarks
1. An urban water management program called Sponge City (SPC) is put forward in China in 2014 in order to relieve the flood inundation and water shortage situation. What is the present situation in the country level about construction, investments, local programs, etc.?
2. The concept of a Sponge City (SPC) is almost similar to the LIDs (Low Impact Developments) of USA, the SuDs (Sustainable Drainage Systems) of United Kingdom and the WSUDs (Water Sensitive Urban Design) of Australia, which exist some decades ago. What is the Chinese interpretation and differences of the SPC concept concerning LIDs, SuDs and WSUDs?
3. I believe that the manuscript will be more attractive for the international reader if a short text about the Sponge City (SPC) project be added.
4. Some researchers have experience on the Sponge City (SPC) project. I propose very kindly, to the authors to take into account the following interesting citations.
Li, X.; Li, J.; Fang, X.; Gong, Y.; Wang, W. Case studies of the sponge city program in China. World Environ. Water Resour. Congr. 2016.
Cheng, D.-X.; Liu, Y.; Chen, J. Summary of Sponge Road Research. International Conference on Manufacturing Construction and Energy Engineering (MCEE) 2016. ISBN: 978-1-60595-374-8.
Li, H.; Ding, L.; Ren, M.; Li, C.;Wang, H. Sponge City Construction in China: A Survey of the Challenges and Opportunities. Water 2017, 9, 594
Zevenbergen, C.; Fu, D.; Pathirana, A. Transitioning to Sponge Cities: Challenges and Opportunities to Address Urban Water Problems in China. Water 2018, 10, 1230; doi:10.3390/w10091230
Nie, L.-M.; Jia H.-F. China’s Sponge City Construction: Ambition and challenges. VANN 2018, 02, 159-167.
Specific remarks
Line 16
“….and Storm Water Management Model (SWMM). The performance…”
should be
“….and Storm Water Management Model (SWMM) of USEPA. The performance…”
Line 42
“Bai et, al. [14]….”
should be
“Bai et al. [19]….”
Line 69”
“P.M.Avellaneda et al. [24] showed…”
should be
“Avellaneda et al. [24] showed…”
Lines 85, 86, 87, 89, 90, 91, 93, 97, 98
Authors state elsewhere “Research Area” (Line 85), elsewhere “study catchment” (Lines 86, 91, 93), elsewhere “study area” (Lines 87, 97, 98) and elsewhere “research catchment” (Lines 89, 90). Please, keep one term throughout the main text.
Lines 97 & 98
“Figure 1. Study area of Sucheng district: (a) Land use of the study area; (b) Diagram of the pipe networks for the study area; (c) Generalized sketch map of sub-catchments.”
should be
“Figure 1. Study area of Sucheng district: (a) Land use; (b) Diagram of the pipe network; (c) Generalized sketch map of sub-catchments.”
Line 101
“…recommended by the model manual [14]”
should be
“…recommended in the SWMM manual [14]”.
Lines 101 & 102
“The manning roughness coefficient for the impervious area (N-Imperv) is 0.012, and the manning roughness coefficient for the pervious area (N-Perv) is 0.1.”
should be
“The Manning roughness coefficients for the impervious area (N-Imperv) and the pervious area (N-Perv) are 0.012 and 0.1 respectively.”
Table 1, Line 3, Column 4
Please explain the symbol “n” in the main text.
Lines 100-106
Authors state:
“Water quantity parameters of sub-catchments are selected based on empirical values recommended by the model manual [14]. The manning roughness coefficient for the impervious area (N-Imperv) is 0.012, and the manning roughness coefficient for the pervious area (N-Perv) is 0.1. The depth of depression storage on the impervious area (Destore-Imperv) is 3.2 mm, and on the pervious area (Destore-Perv) is 6.6 mm. We choose the Horton model as the infiltration method. The maximum infiltration rate is 75 mm/h, while minimum is 3.81 mm/h. Partial parameters of each LID technique are shown in Tables 1-4.”
COMMENTS:
a. The selection of quantity parameters is unclear. Reader is not able to understand how these parameters are selected. For example, the SWMM manual [14] define for N-Imperv a range of values 0.006-0.05, for N-Perv 0.08-0.5, for Destore-Imperv 0.2-5.0, for Destore-Perv. 2-10 and so on. Please, combine your selections for N-Imperv, N-Perv, etc. with the findings of a recent article (see reference No. 14): “Bai, Y.R.; Zhao, N.; Zhang, R.Y.; Zeng, X.F. (2019) - Storm Water Management of Low Impact Development in Urban Areas Based on SWMM. Water., 11. doi:10.3390”.
b. Please, explain precisely how do you select the values of the N-Imperv, N-Perv, Destore-Imperv and so on, on the basis of which criteria do you use.
c. General speaking, the unity "2.2 Determination of Parameters" should be rewritten in a manner clear and in detail.
Lines 114 & 115
Authors state: “Each LID facility was distributed based on practical experience and related references [15]”.
COMMENTS:
a. This sentence concerning the distribution of LID facility is obscure. Please, clarify the “related references”. What do you mean?
According to the manuscript, the related reference is only the No. 15 ("15. Rodriguez Droguett, B. Sustainability assessment of green infrastructure practices for stormwater management: A comparative emergy analysis (M.S.),2011"), which is a M.Sc. of 2011 and the retrieval from the https://search.proquest.com/docview/900864997/abstract/6E6468B55EF4 you give the possibility to download only the "INTRODUCTION" (16 pp.)!
b. Please, clarify the term "practical experience". What does it mean? Do authors mention in their practical experience or in general?
Table 5
The selection of scenarios in unclear. Reader is not able to understand what is included in different scenarios. For example, what does it mean “A*(0~100%, Interval of 20%)” in scenario 1, “B*(0~100%, Interval of 20%)” in scenario 2, “C*(0~100%, Interval of 20%)” in scenario 3 and “D*(0~100%, Interval of 20%)” in scenario 4.
Lines 128, 132 & 136
Authors state elsewhere “return period, P” (see Lines 128 & 132) and elsewhere “rainfall period” (see Line 136, legend of Figure 3). Please, keep one term throughout the main text.
Line 134
“The rainfall hydrograph is shown in Figure 3”
should be
“The design hyetograph developed for the study area from equation (1) and return period 80% is shown in Figure 3.”
Line 136, Legend of Figure 3
According to my opinion, Figure 3 shows a “design hyetograph” and no a “rainfall hydrograph”
Lines 142 & 143
“….which is described in Equation (2) and Equation (3)”
should be
“….which is described in Equations (2) and (3)”
Line 152
“…and peak reduction (0.333) [32]. The…”
should be
“…and peak reduction (0.333) [6]. The…”
Line 154
“…measures in Suqian city are shown in Table 6 [27] [33]…”
should be
“…measures in Suqian city are shown in Table 6 [27, 32]…”
Tables 7 & 8; Line 1, Column 1
“LID proportion” should be “LID proportion of maximum construction area”
Line 171
“…were presented in Table 7, Table 8 and Figure 5.”
should be
“…were presented in Tables 7, 8 and Figure 5.”
Tables 7 & 8
Please, clarify the term “difference value”. What does it mean?
Figure 5
(a) y-axis: “Reduction rate / ha” should be “Reduction rate / ha (%)”
(b) Check please, in y-axis: Are “0.50%, 0.40%, 0.30%, 0.20%” correct?
Line 194
“…with the results of Nan Li et al [34].”
should be
“…with the results of Nan Li et al. [33].”
Line 197
“…and relevant literatures [33].”
should be
“…and relevant literatures [32].”
Lines 234 & 235
“With the increasing area of LID facilities, LID facilities per unit area perform better in runoff reduction”.
COMMENT: This sentence is unclear and it has to be rewritten correctly in correct English.
Lines 236 & 237
“Different LID facilities have different performances in the reduction rate of runoff volume and peak flow, permeable pavements perform...”
should be
“Different LID facilities have different performances in the reduction rate of runoff volume and peak flow. Specifically, permeable pavements perform...”
Line 260
“Paule-Mercado, M.A.; Lee, B.Y.; Memon, S.A.; Umer, S.R.; Salim, I.”
should be
“Paule-Mercado, M.A.; Lee, B.Y.; Memon, S.A.; Umer, S.R.; Salim, I.; Lee, C.-H.”
Line 267
“Li, Q.; Wang, F. “
should be
“Li, Q.; Wang, F.; Yu, Y.; Huang, Z.; Li, M; Guan Y.”
Line 302
“Luan, Q.H.; Fu, X.R.”
should be
“Luan, Q.H.; Fu, X.R.; Song, C.P.; Wang, H.C.; Liu, J.H.;Ying Wang Y.”
Line 305
“Avellaneda, P. M.; Jefferson, A. J.”
should be
“Avellaneda, P. M.; Jefferson, A. J.; Grieser, J. M.; Bush, S. A.”
Line 320
Please, delete reference No. 32 “Li, Q.; Wang, F.“; renumber the references thereafter; correct the main text.
COMMENT: References No 6 and 32 are same!
Author Response
Dear professor,
Greetings.
We are writing to answer the questions that you have proposed in this paper. Thank you very much for your time and insightful suggestions. All the comments and suggestions are valuable. We think it’s very necessary to improve these deficiencies.
We added the Sponge City (SPC)、Low Impact Development (LID)、Sustainable Drainage Systems (SuDs)、Water Sensitive Urban Design (WSUDs) and Green Stormwater infrastructure (GSI) in the introduction. For SPC and LID, we make more detailed introduction. We were very sorry not to refer to these literatures before [1-5]. We read these literatures very carefully and found all the literatures were very valuable for this paper. So, we cited them in the paper. As shown in the introduction.
In order to keep one term through the main text, we changed terms “Research Area”, “Study catchment” and “Research catchment” into “Study area”.
Table 1, Line 3, Column 4
We modified “n” into “roughness”.
2.2 Determination of Parameters
“Water quantity simulation is related to underlying surfaces, and parameters in SWMM are very important to the accuracy of simulation results [20]. Initial water quantity parameters were selected based on empirical values recommend in the SWMM manual. Then, parameters were calibrated and validated based on the comprehensive runoff coefficient (CRC) method [29]. The manning roughness coefficient for the impervious area (N-Imperv) and the pervious area (N-Perv) are 0.012 and 0.1 respectively; the depth of depression storage on the impervious area (Destore-Imperv) is 3.2 mm, and on the pervious area (Destore-Perv) is 6.6 mm. We chose the Horton model as the infiltration method. The maximum infiltration rate is 75 mm/h, while minimum is 3.81 mm/h.”
2.3 Design Scenarios
“In scenario 1, the area of green roofs increases at an interval of 20% from 0% to 100%, while the areas of the other LID facilities occupy 40% of the corresponding maximum construction area. In scenario 2, the area of permeable pavements increases at an interval of 20% from 0% to 100%, while the areas of the other LID facilities occupy 40% of the corresponding maximum construction area. In scenario 3, the area of concave greenbelts increases at an interval of 20% from 0% to 100%, while the areas of the other LID facilities occupy 40% of the corresponding maximum construction area. In scenario 4, the area of rain gardens increases at an interval of 20% from 0% to 100%, while the areas of the other LID facilities occupy 40% of the corresponding maximum construction area. A (83.7742 ha) is the maximum construction area for green roofs, B (45.2075 ha) is the maximum construction area for permeable pavements, C (61.1726 ha) is the maximum construction area for concave greenbelts, D (50.0771 ha) is the maximum construction area for rain gardens. Proportion of maximum construction area 40% is determined based on the findings of Zhang et al. [40].” In addition, “A*(0~100%, Interval of 20%)”, “B*(0~100%, Interval of 20%)”, “C*(0~100%, Interval of 20%)”, and “D*(0~100%, Interval of 20%)” were modified as “A* (0%, 20%, 40%, 60%, 80%, 100%)”, B* (0%, 20%, 40%, 60%, 80%, 100%)”, “C* (0%, 20%, 40%, 60%, 80%, 100%)”, and “D* (0%, 20%, 40%, 60%, 80%, 100%)”.
In order to keep one term through the main text, we changed term “rainfall period” into “return period”.
D-value (different value) refers to the difference values between 0% and 20%, 20% and 40%, 40% and 60%, 60% and 80%, 80% and 100% in each scenario.
Figure 5. y-axis
Modified as “0.50, 0.40, 0.30, 0.20”.
“With the increasing area of LID facilities, LID facilities per unit area perform better in runoff reduction”.
Modified as “as the LID proportion of maximum construction area becomes smaller, the capacity for stormwater management of each LID facility is limited.”
Reference No 5 has been replaced by References No 29 and 39.
References No 6 and 32 are same.
We have deleted Reference 32. The structure of the references has also been adjusted.
For all recommended details such as, “…” should be “…”
All of these have been modified as your suggestions.
Thank you very much for your insightful comments and construction suggestions. We think these comments and suggestions improved the manuscript greatly.
Best regards to you.
2019/1/29.
[1] Li, X.; Li, J.; Fang, X.; Gong, Y.; Wang, W. Case studies of the sponge city program in China. World Environ. Water Resour. Congr. 2016.
[2] Cheng, D.-X.; Liu, Y.; Chen, J. Summary of Sponge Road Research. International Conference on Manufacturing Construction and Energy Engineering (MCEE) 2016. ISBN: 978-1-60595-374-8.
[3] Li, H.; Ding, L.; Ren, M.; Li, C.;Wang, H. Sponge City Construction in China: A Survey of the Challenges and Opportunities. Water 2017, 9, 594
[4] Zevenbergen, C.; Fu, D.; Pathirana, A. Transitioning to Sponge Cities: Challenges and Opportunities to Address Urban Water Problems in China. Water 2018, 10, 1230; doi:10.3390/w10091230
[5] Nie, L.-M.; Jia H.-F. China’s Sponge City Construction: Ambition and challenges. VANN 2018, 02, 159-167.

Reviewer 2 Report
This is an interesting and important topic. The authors have done a great job applying several established models in a real-world setting. However, I sense a lack of uniqueness in research design/methodology. I am totally convinced how this study contributes to existing literature. The authors need to clarify this in the beginning and at the end of the paper. The introduction needs to be re-worked by highlighting more existing research that has followed similar or same tools or methods. In the end, authors should compare their findings with other similar studies. Has rain garden produced a similar type of results in other similar studies done in other areas?
Green infrastructure needs to be more defined or explained in the beginning.
The SWMM model is an EPA model. It has been mentioned as EAP (page 2). Please correct it and also say the full name of the organization – The U.S. Environmental Protection Agency.
The authors used past tense and present tense at the same time. For example, on page 3, the authors wrote “The study catchment area was …” and then they wrote, “.. the study catchment is….” Please be consistent with your tense. I think everything that you “did” should be written in past tense.
The legends in Figure 1 are too small and therefore difficult to read.
Suddenly from section 2.2, the write-up became too technical and any reader without prior experience with SWMM model would have a hard time understanding a number of things being described.
The introductory paragraph under section 3 is unnecessary because authors have already discussed the methods in section 2. Figure 4 should be part of the methodology, not results.
The discussion should be around the implication of this study. More importantly, would this finding be similar in other areas or is it relevant in only this case?
The authors have talked about economic and environmental benefits and their study completely ignores the social aspect of green stormwater infrastructure. Why the social factors were not considered? The authors need to address this issue. At a minimum, they need to include this as a limitation of the study. For reference, please check out the following articles:
Christman, Z.; Meenar, M.; Mandarano, L.; Hearing, K. Prioritizing Suitable Locations for Green Stormwater Infrastructure Based on Social Factors in Philadelphia. Land 2018, 7, 145.
Mandarano, L.; Meenar, M. Equitable Distribution of Green Stormwater Infrastructure: A Capacity-Based Framework for Implementation in Disadvantaged Communities. Local Environment 2017, 22(11), 1338-1357.
Author Response
Dear professor,
Greetings.
We are writing to answer the questions that you have proposed in this paper. We are very grateful for your insightful comments and construction suggestions. We think it’s very necessary to improve these deficiencies.
We added the Sponge City (SPC)、Low Impact Development (LID)、Sustainable Drainage Systems (SuDs)、Water Sensitive Urban Design (WSUDs), Green Stormwater infrastructure (GSI) and the U.S. Environmental Protection Agency (EPA) Storm Water Management Model (SWMM) in the introduction. We provided greater details for SPC and LID. We were very sorry not to refer to these literatures before [1-2]. We read these two literatures very carefully and found all the literatures were very valuable for this paper. So, we cited them in the paper. In addition, many relevant literatures were added in it.
We have corrected SWMM as the U.S. Environmental Protection Agency (EPA) Storm Water Management Model (SWMM).
We checked the tenses and corrected them.
The legends in Figure 1 have been enlarged so as to be read clearly.
2.2 Determination of Parameters
“Water quantity simulation is related to underlying surfaces, and parameters in SWMM are very important to the accuracy of simulation results [20]. Initial water quantity parameters were selected based on empirical values recommend in the SWMM manual. Then, parameters were calibrated and validated based on the comprehensive runoff coefficient (CRC) method [29]. The manning roughness coefficient for the impervious area (N-Imperv) and the pervious area (N-Perv) are 0.012 and 0.1 respectively; the depth of depression storage on the impervious area (Destore-Imperv) is 3.2 mm, and on the pervious area (Destore-Perv) is 6.6 mm. We chose the Horton model as the infiltration method. The maximum infiltration rate is 75 mm/h, while minimum is 3.81 mm/h.”
The introductory paragraph under section 3 has been deleted. Figure 4 (Figure 1) has been placed in the part of methodology.
Some findings are similar in other areas but some results are just relevant in this case. We discussed it in the section of Results and Discussion.
Considering that social benefits might be difficult to be estimated accurately because of the limited data, we selected environmental benefits and economic benefits in this research. In addition, we included this as a limitation of the study in the section of Conclusion.
Thank you very much for your insightful comments and construction suggestions. We think these comments and suggestions improved the manuscript greatly.
Best regards to you.
2019/1/29.
[1] Christman, Z.; Meenar, M.; Mandarano, L.; Hearing, K. Prioritizing Suitable Locations for Green Stormwater Infrastructure Based on Social Factors in Philadelphia. Land 2018, 7, 145.
[2] Mandarano, L.; Meenar, M. Equitable Distribution of Green Stormwater Infrastructure: A Capacity-Based Framework for Implementation in Disadvantaged Communities. Local Environment 2017, 22(11), 1338-1357.

Reviewer 3 Report
Please see the attachment.

Author Response
Dear professor,
Greetings.
We are writing to answer the questions that you have proposed in this paper. We are very grateful for your insightful comments and construction suggestions. We think it’s very necessary to improve these deficiencies. Thanks for your time.
Introduction:
1, China has experienced severe flood disasters because of high-speed urbanization and extreme climate events. Therefore, it’s of great significance for China to propose new suitable policies to deal with these hazards.
2, Chinese government launched sponge city construction to address these challenges. We provided greater details for Sponge City (SPC) and Low Impact Development (LID). The concept of SPC was almost similar to the the SuDs (Sustainable Drainage Systems), the WSUDs (Water Sensitive Urban Design), Water Sensitive Urban Design (WSUDs), Green Stormwater infrastructure (GSI). These urban water terms have also been added in the introduction.
3, The U.S. Environmental Protection Agency (EPA) Storm Water Management Model (SWMM) and Analytic Hierarchy Process (AHP) method were used to simulate the rainfall-runoff process.
4, The simulation results can provide useful guidance for the selection of LID techniques, and can also provide technical support for sponge city construction in urban areas. Some suggestions for sponge city construction are given in the section of conclusions.
In addition, some relevant literatures have been added to provide the research background.
Material and Methods:
We modified the study area in Section 2.1
The main reasons were that two storms (the storm in September 17, 2010; the storm in June 23, 2016) in the study area had caused severe economic losses to residents.
The legends in Figure 2 (Figure 1) have been enlarged so as to be read clearly. Figure 2(a) (From the Google map) has been added in order to show the current conditions of the Sucheng district.
In order to analyze the hydrological performance of each LID technique and obtain the optimal proportion of the LID technique, four LID scenarios were designed based on the research of Zhang et al. [40].
In order to show the distribution of each LID facility, Figure 3 (Figure 2) was placed in there. We haven't found a better way to draw it yet. We tried to draw the LID facilities on the map but it wasn’t worked very well. We were very sorry for it.
Results and Discussion
Some questionnaires of sponge city construction have been employed by relevant government departments. These questionnaires are not available to the public. We modified as “Costs of LID facilities were determined based on the local price level and the researches of Zhang et al. [40] and Tang et al. [45].”
Figure 5(b-e) were added to make the results and discussion more convictive. We also compared the findings of other similar studies.
Conclusions
Some suggestions for sponge city construction were given in the section of conclusions, and the limitation of this study was concluded in order to provide some suggestions for the future research.
Thank you very much for your insightful comments and construction suggestions. We think these comments and suggestions improved the manuscript greatly.
Best regards to you.
2019/1/29.

Round 2
Reviewer 1 Report
This version of the manuscript (round 2) is far better from the original (round 1). However, it needs some corrections yet (please see my comments) in order to be suitable for publish.
Moreover, the manuscript has to be corrected in English language.
Specific remarks
References from No. 24 “Dietz…” until No.29 “Bai …” have to be renumbered according to the main text.
Lines 12 & 13
“Sponge city construction can effectively mitigate urban flooding and improve the urban rainwater utilization, Low Impact Development (LID) techniques are often applied in it.”
COMMENT: This sentence has to be written in correct English.
Line 70
“Bai et al. [29] found that LID facilities….”
should be
“Bai et al. [23] found that LID facilities….”
COMMENT
According to instructions for authors of WATER: “References must be numbered in order of appearance in the text (including table captions and figure legends) and listed individually at the end of the manuscript”.
Lines 71 & 72
“….on storage. Besides that, there are many researches on LID techniques.”
should be
“….on storage. It is noted, there are many researches on LID techniques.”
or alternatively (preferable)
“….on storage. Besides that, there are many researches on LID techniques.”
Line 76
“Alfredo et al. [23] reported…” should be “Alfredo et al. [24] reported…”
Line 77
“Dietz’s [24] experiment results…” should be “Dietz’s [25] experiment results…”
Line 81
“…improving the environment [25]. Permeable…” should be “…improving the environment [26]. Permeable…”
Line 83
“….in the soil [26-27]. They are…” should be “….in the soil [27-28]. They are…”
Line 88
“Abbott and Comino-Mateos [28] found…” should be “Abbott and Comino-Mateos [29] found…”
Line 92
“…areas [29]. Luan et al. [30] evaluated…”
should be
“…areas [23]. Luan et al. [30] evaluated…”
Line 105
“…simulation [8] [13] [20] [29]. SWMM…” should be “…simulation [8] [13] [20] [23]. SWMM…”
Line 128
“…a total area of 270 hectares. Sucheng…”
should be
“…a total area of 270 ha. Sucheng…”
Line 151
“…runoff coefficient (CRC) method [29]. The manning roughness…”
should be
“…runoff coefficient (CRC) method [23]. The manning roughness…”
Lines 151 & 152
“The manning roughness coefficient for the impervious area (N-Imperv) and the pervious area (N-Perv) are…”
should be
“The manning roughness coefficients for the impervious area (N-Imperv) and the pervious area (N-Perv) are…”
Lines 157 & 158
“….proposed by Bai et al. [29] and Huang et al. [39].”
should be
“….proposed by Bai et al. [23] and Huang et al. [39].”
Line 172
“….and Bai et al. [29]. Green roofs,…”
should be
….and Bai et al. [23]. Green roofs,…”
Table 7, Column 1, Lines 3, 5, 7, 9
“D-value” should be “D-value*”
Line 237
“D-value refers to…” should be “*D-value refers to…”
Table 8, Column 1, Lines 3, 5, 7, 9
“D-value” should be “D-value*”
Line 247
“D-value refers to…” should be “*D-value refers to…”
Line 300
“….facilities per hectare were ranked …” should be “….facilities per unit area (ha) were ranked …”
Lines 301-303
“Li et al. [13] obtained similar results in the research: the runoff control efficiency of the concave greenbelt (about 1 ha) might be greater than that of the rain garden (about 1 ha).”
COMMENT: This sentence is unclear. It has to be written in correct English.
Author Response
Dear professor,
Greetings.
We are writing to answer the questions that you have proposed in this paper. We are very grateful for your insightful comments and construction suggestions. We think it’s very necessary to improve these deficiencies. Thanks for your time.
References from No. 24 “Dietz…” until No.29 “Bai …” have been renumbered according to the main text. thanks for your reminder.
Lines 12 & 13 & 14
“Sponge city construction can effectively mitigate urban flooding and improve the urban rainwater utilization, Low Impact Development (LID) techniques are often applied in it.”
We modified it as “Sponge city construction can effectively mitigate urban flooding and improve the urban rainwater utilization. Low Impact Development (LID) is regarded as a sustainable solution for urban stormwater management.”
Lines 298 & 300
“Li et al. [13] obtained similar results in the research: the runoff control efficiency of the concave greenbelt (about 1 ha) might be greater than that of the rain garden (about 1 ha).”
We modified it as “Li et al. [13] obtained similar results in their research: concave greenbelts performed better in runoff reduction than rain gardens.”
We have checked the paper for grammar.
For all recommended details such as, “…” should be “…”
All of them have been modified as your suggestions.
Thank you very much for your insightful comments and construction suggestions. Thank you for your careful review of this paper. And thanks for your time.
Best regards to you.
2019/2/11.

Reviewer 2 Report
Thanks to the authors to revise the manuscript. This version looks much better.
Author Response
Thanks for your time.
Best regards to you!
2019/2/11.
Reviewer 3 Report
The manuscript has been greatly improved.
Author Response

(The authors gave the same response as above.)
